# Immunotherapy in Early-Stage Non-Small Cell Lung Cancer (NSCLC): Current Evidence and Perspectives

Chiara Lazzari [1], Calogera Claudia Spagnolo [2], Giuliana Ciappina [2], Martina Di Pietro [2], Andrea Squeri [2], Maria Ilenia Passalacqua [2], Silvia Marchesi [3], Vanesa Gregorc [1,†] and Mariacarmela Santarpia [2,*,†]

1   Candiolo Cancer Institute, Fondazione del Piemonte per l'Oncologia (FPO)-IRCCS, 10060 Torino, Italy
2   Medical Oncology Unit, Department of Human Pathology "G.Barresi", University of Messina, 98158 Messina, Italy
3   Fondazione IRCCS Istituto Nazionale dei Tumori, 20133 Milano, Italy
*   Correspondence: mariacarmela.santarpia@unime.it
†   These authors contributed equally to this work.

**Abstract:** Lung cancer is the leading cause of cancer deaths in the world. Surgery is the most potentially curative therapeutic option for patients with early-stage non-small cell lung cancer (NSCLC). The five-year survival for these patients remains poor and variable, depending on the stage of disease at diagnosis, and the risk of recurrence following tumor resection is high. During the last 20 years, there has been a modest improvement in the therapeutic strategies for resectable NSCLC. Immune checkpoint inhibitors (ICIs), alone or in combination with chemotherapy, have become the cornerstone for the treatment of metastatic NSCLC patients. Recently, their clinical development has been shifted in the neoadjuvant and adjuvant settings where they have demonstrated remarkable efficacy, leading to improved clinical outcomes. Based on the positive results from phase III trials, ICIs have become a therapeutic option in neoadjuvant and adjuvant settings. On October 2021 the Food and Drug Administration (FDA) approved atezolizumab as an adjuvant treatment following surgery and platinum-based chemotherapy for patients with NSCLC whose tumors express PD-L1 ≥ 1%. In March 2022, nivolumab in combination with platinum-doublet chemotherapy was approved for adult patients with resectable NSCLC in the neoadjuvant setting. The current review provides an updated overview of the clinical trials exploring the role of immunotherapy in patients with early-stage NSCLC, focusing on the biological rationale for their use in the perioperative setting. We will also discuss the role of potential predictive biomarkers to personalize therapy and optimize the incorporation of immunotherapy into the multimodality management of stage I-III NSCLC.

**Keywords:** immunotherapy; immune checkpoint inhibitors; lung cancer; NSCLC; early stage; adjuvant; neoadjuvant; predictive biomarkers

## 1. Introduction

Lung cancer is the leading cause of cancer deaths in the world [1]. Non-small cell lung cancer (NSCLC) accounts for 80% of cases. Surgery is the most potentially curative therapeutic option, and the treatment of choice in patients with stage I and II cancers, followed by adjuvant platinum-doublet chemotherapy in those with stage II cancers [2]. Despite the role of adjuvant therapy in reducing the risk of recurrence by eliminating systemic, micrometastatic disease, its impact on overall survival (OS) remains modest. The LACE meta-analysis, including data from 4584 patients enrolled in five randomized trials evaluating the efficacy of postoperative platinum-based chemotherapy, showed a five-year benefit of only 5.4%. The advantage was observed in patients with stage II and III (HR = 0.83; CI, 0.73 to 0.95), but not those with stage I (HR = 0.93; CI, 0.78 to 1.10) [2]. Multimodality approaches, including chemotherapy, radiotherapy and surgery have represented the cornerstone in patients with stage IIIA. Despite these attempts, the five-year survival of resectable NSCLC patients ranges from 67% for those with T1N0 (IA)

disease to 23% for those with T1-3N2 (IIIA), and the risk of recurrence following tumor resection remains high [3]. Data from the literature indicate that approximately 20–30% of NSCLC patients with stage I, 50% of those with stage II and 60% of those with stage IIIA die within five years [4].

Different preoperative strategies with neoadjuvant therapies have been explored with the aims to downstage the tumor before surgery, allow the use of minimally invasive surgery, inhibit the early development of micro-metastases, thus reducing the incidence of systemic relapse, and improve patients' survival. These approaches have been associated with limited efficacy. The phase III North American Intergroup 0139 trial evaluated the impact of surgery in 396 patients with T1-3 N2 disease who received radiotherapy up to 45 Gy, concurrent with two cycles of cisplatin and etoposide [5]. In the absence of progression, patients were randomized between surgery and the continuation of radiotherapy to 61 Gy. The primary endpoint was OS. Results showed that surgery determines a prolongation of progression-free survival (PFS) with no OS improvement. The OS was significantly improved in the subgroup that underwent lobectomy, while worse OS was observed in those receiving pneumonectomy. A meta-analysis conducted by the NSCLC Meta-Analysis Collaborative Group involving 15 randomized clinical trials including patients with stage IB to IIIA disease compared chemotherapy with subsequent surgery versus surgery alone demonstrated an absolute 5% survival benefit at 5 years with neoadjuvant chemotherapy [6]. These data suggest the need to develop more effective strategies to reduce the risk of recurrence and improve the survival of resectable NSCLC.

The current review provides an overview of the rationale for the design of clinical trials exploring the efficacy of immune checkpoint inhibitors (ICIs) in the neoadjuvant and adjuvant settings.

## 2. The Immunological Bases for the Use of ICIs in the Preoperative Setting

Preoperative treatment offers the opportunity to study *in vivo* radiological and adaptive responses of tumors to systemic therapy. As a consequence, it can be potentially used to identify prognostic and predictive factors to tailor subsequent adjuvant treatment strategies [7].

Preclinical data from syngeneic mice models of NSCLC showed that the administration of three doses of neoadjuvant nivolumab or ipilimumab + nivolumab significantly prolonged survival over three doses of adjuvant nivolumab or ipilimumab + nivolumab ($p < 0.05$). The greatest number of tumor-infiltrating lymphocytes (TILs) and highest CD8+ TIL density found in resected primary tumors from mice receiving neoadjuvant ICIs represent the immunological bases for the major benefit observed with neoadjuvant strategies [8]. When ICIs are used in the neoadjuvant setting, due to the presence of the primary tumor, there is a higher probability of inducing tumor-specific CD8+ T cells and peripheral tumor-specific immune responses. Once activated, CD8+ T cells circulate into the blood, where they expand and infiltrate the organs. The T cell response favors the release of tumor antigens, which are recognized by the antigen-presenting cells (APC), therefore inducing the activation of prime naïve T cells. As a consequence, micrometastatic lesions are destroyed and a stable pool of CD8+ T cells remains, thus maintaining the T cells' response and reducing the risk of recurrence (Figure 1). These represent the bases leading to prolonged survival.

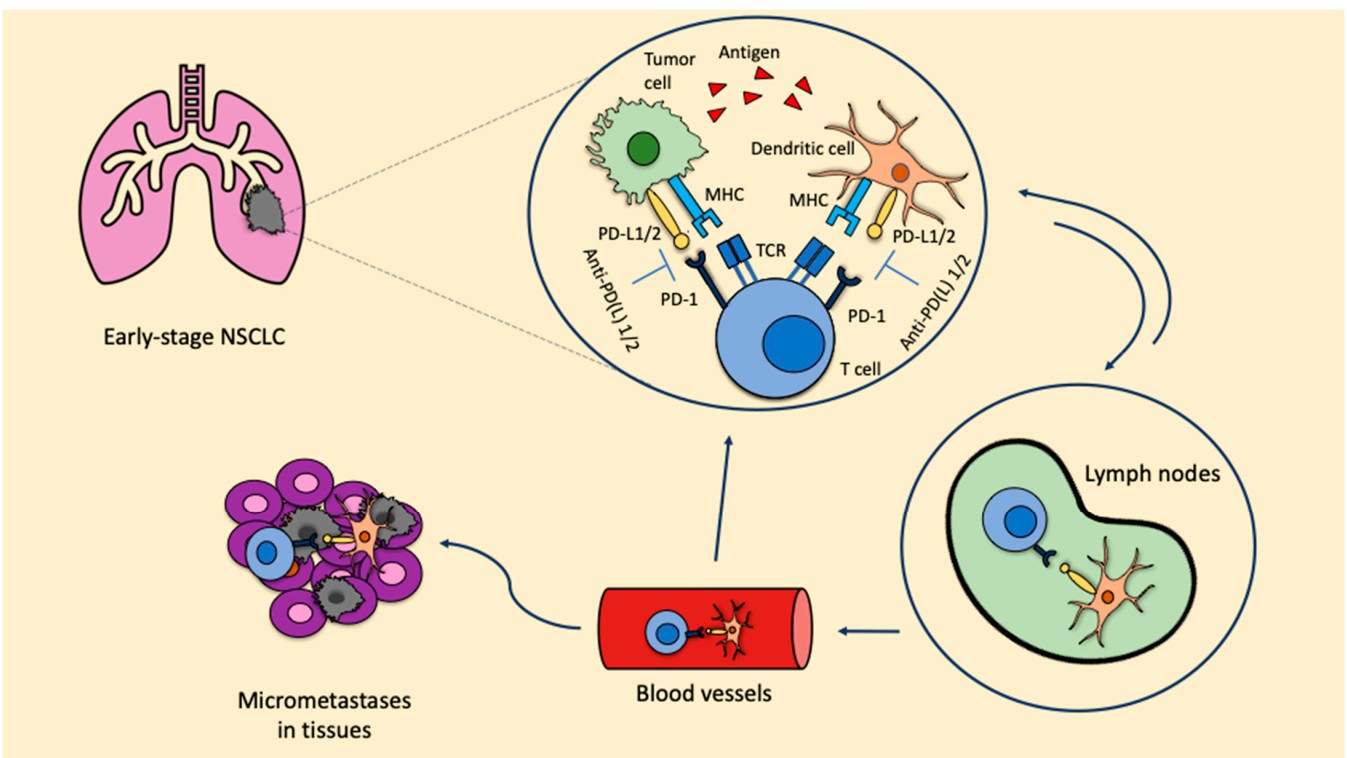

**Figure 1.** ICIs in the neoadjuvant setting: biological rationale. Anti-PD-L1: anti-programmed cell death-1; MHC: major histocompatibility complex; NSCLC: non-small cell lung cancer; PD-1: programmed cell death protein; PD-L1/2: programmed death ligand 1/2; TCR: T cell receptor.

The inhibition of the PD-1/PD-L1 axis allows T cells to kill tumoral cells and also induces the expansion of tumor-specific T cells in the tumor microenvironment. This expansion is mainly led by PD-L1/2 expressing dendritic cells in the tumor. Furthermore, dendritic cells that contain tumor antigens shift to lymph nodes, where they present these antigens to tumor-specific T cells, enhancing the productive stimulation of these. To this point, activated tumor-specific T cells can enter into the blood circulation and reach micrometastases in the tissues, starting a series of specific and durable antitumor immune responses. Some of these tumor-specific T cells return through blood vessels to the primary tumor where they can potentiate the antitumoral activity.

### 3. Efficacy of Monotherapy with PD-1 and PD-L1 Inhibitors in the Neoadjuvant Setting

In 2018, a pilot study evaluated the safety and feasibility of the anti-PD-1 agent nivolumab, administered as a neoadjuvant therapy for two cycles in 21 patients with resectable stage I-IIIA NSCLC [9]. Secondary endpoints included radiologic and pathological responses. Feasibility was defined as the delay of surgery by no more than 37 days following nivolumab. Results demonstrated that nivolumab did not delay surgery, as the median interval between the last cycle and surgery was 18 days. Only one patient did not complete the two cycles of pre-planned therapy due to the onset of grade 3 pneumonia, which did not preclude surgery. Despite partial radiological responses being observed only in 10% of cases, and 86% developing stable disease, pathologic down-staging was registered in 40% of patients. Major pathological responses (MPR), defined as ≤10% residual viable tumor cells in the primary tumor and lymph nodes, were found in 45% of cases, while pathological complete response (pCR) in the primary tumor, defined as 0% of residual viable tumor cells, was observed in three patients. The median degree of pathological regression was—65%. In the resected tumor and in the peripheral blood, the expansion of CD8+ T cells was evidenced.

Similar findings were observed in the phase II LCMC3 trial [10], which included 181 resectable NSCLC patients with stage IB-IIIA and selected IIIB (T3N2 or T4) who received two cycles of neoadjuvant atezolizumab. The primary endpoint was the percentage of MPR. Eighty-eight percent of the patients underwent surgery. The MPR rate was 20% and the pCR rate was 6%. The lower percentage of MPR registered in the LCMC3 trial in comparison to that observed in the study exploring nivolumab might be related to the large number of patients with stage III being included (51% in the LCMC3 study vs. 33%). The PD-L1 levels were significantly correlated with pathologic response ($p < 0.001$). Among the enrolled patients, ten carried EGFR-activating mutations and six carried EML4-ALK translocations. No radiological response or MPR was registered. Lesser pathologic responses were identified in patients harboring STK11 mutations compared with wild-type. Encouraging 3-year disease-free survival (DFS) and OS of 72% and 80%, respectively, were reported. A significant expansion of peripheral blood-activated CD8+ T cells was observed in patients obtaining tumor regression.

The phase II IFCT-IONESCO study was designed to evaluate the feasibility of the anti-PD-L1 agent durvalumab as a neoadjuvant therapy in 46 resectable patients with stage IB-IIIA NSCLC [11]. The primary endpoint was complete surgical resection, while secondary endpoints included the interval between the first administration of durvalumab and therapy, the tumor response, the rate of MPR, DFS, OS and safety. Eighty-nine percent of the patients underwent complete resection, with a time interval between the beginning of therapy and surgery of 37 days. The MPR was 19%. No correlation was observed between PD-L1 expression and the percentage of MPR. Different from the other trials, the mortality rate at 90 days following surgery was high, with 9% of deaths due to postoperative complications. However, three out of four patients who died had cardiovascular comorbidities.

The phase II NEOMUN study explored the safety of two cycles of another monoclonal antibody targeting PD-1, pembrolizumab, when administered as a neoadjuvant treatment in resectable NSCLC patients with stage IIA-IIIA cancers [12]. Co-primary endpoints were the percentage of adverse events (AEs), the overall response rate (ORR) according to RECIST and iRECIST criteria, the percentage of pathological responses and the functional PET activity. Secondary endpoints included DFS and OS. Preliminary results showed MPR in 27% of patients, and treatment was associated with a good tolerability profile [13].

All these data indicate that treatment with ICIs in the neoadjuvant setting is associated with a high percentage of resection rates, a good safety profile and a significant probability of prolonging patients' survival. Data from the literature indicate that intratumoral immune cell infiltration, as a consequence of the treatment with ICIs, is associated with pseudo-progression in approximately 0.6–5.8% of patients with NSCLC [14,15]. Nodal immune flare, registered in 13–19% of patients, refers to the presence, at the histological evaluation, of non-caseating granulomas and the absence of tumor cells within those lymph nodes radiologically suspected for tumor progression [16]. In the neoadjuvant setting, the identification of the nodal immune flare phenomenon is particularly critical in order to differentiate between true progression and pseudo-progression, to not preclude curative surgery and to avoid an inappropriately large radiation field. Furthermore, in 7% of cases receiving neoadjuvant ICIs and no evidence of radiological nodal progression, non-caseating granulomas have been documented [16]. Macrophages, dendritic cells, cytotoxic cells, Th1 cells and exhausted CD8+ T cells have been described in the lymph nodes presenting non-caseating granulomas. Moreover, the upregulation of genes associated with the activation of the immune system, including the interferon gamma (IFN-$\gamma$) pathway, and the downregulation of those associated with immune suppression, such as transforming growth factor beta (TGF-$\beta$) and SMAD2/3, have been identified. These data suggest the need to perform a pathological evaluation in the case of suspicious nodal immune flare before defining the therapeutic strategy in patients receiving neoadjuvant treatment with ICIs.

## 4. Combination Strategies in the Neoadjuvant Setting

In order to improve the efficacy of ICIs as a therapeutic strategy in the neoadjuvant setting, different combinatorial treatments have been evaluated, including concurrent chemotherapy and immunotherapy or the combination of dual ICIs.

NEOSTAR was a phase II trial comparing the efficacy of nivolumab to nivolumab + ipilimumab in 44 resectable NSCLC patients with stage IA-IIIA cancers [17]. Nivolumab was administered at the dose of 3 mg/kg every 14 days for three cycles. Those patients randomized in the nivolumab + ipilimumab arm received one cycle of ipilimumab on day 1 at the dose of 1 mg/kg. The primary endpoint was the percentage of MPR. Among the 44 randomized patients, MPR, as assessed by two independent and trained pathologists, was observed in 22% of patients receiving nivolumab and 38% of those receiving nivolumab + ipilimumab. Among the patients enrolled, 37 underwent tumor resection. pCR was observed in 38% of those receiving combination therapy compared to 10% of those in the nivolumab arm. Immune profiling of resected tumor tissues revealed greater tumor infiltration of CD3+, CD3+ CD8+ T lymphocytes and of other immune cell populations, compared to pretreatment tissues, in those patients receiving nivolumab + ipilimumab, thus suggesting that treatment with dual ICIs enhances the immunologic memory. Toxicities were overall manageable, with no new safety concerns compared with the known safety profiles of nivolumab and nivolumab + ipilimumab.

With the aim to increase the percentage of complete pathologic responses and allow curative resection in a higher proportion of patients with stage IIIA NSCLC, the efficacy of ICIs with platinum-doublet chemotherapy was evaluated. The open-label, multicenter, single-arm, phase II NADIM study tested the activity of carboplatin and paclitaxel combined with nivolumab, administered for three cycles, in 46 patients with stage IIIA, resectable NSCLC [18]. Following surgery, patients received adjuvant nivolumab for one year. The primary endpoint was PFS at 24 months. Secondary endpoints included OS at three years, the percentage of ORR and pathological responses, the percentage of patients receiving a complete resection and the safety of the intervention. Prespecified exploratory analyses investigated the predictive role of PD-L1 expression, TMB and immune cell populations in the tumor microenvironment. At 24 months, PFS was 77.1%, while at 36 and 42 months, it was 69.6%, in both cases. The OS at 36 and 42 months was 81.9% and 78.9%, respectively. [19]. Historical data showed that only 30% of patients with stage IIIA NSCLC were alive at three years, thus confirming the significant impact of the addiction of nivolumab on OS prolongation. In this study, neither TMB nor PD-L1 staining predicted long-term survival, while a significant association between ctDNA levels after neoadjuvant chemoimmunotherapy and survival outcomes was demonstrated. According to the RECIST 1.1 criteria, 35 (76%) of 46 patients had an overall response, including 2 (4%) with a complete response and 33 (72%) with a partial response. Eleven (24%) had stable disease and no patients had progressive disease during neoadjuvant therapy. Complete resection was achieved in 89% of cases, with an ORR of 76%. MPR was observed in 83% of patients and pCR in 63% of cases. Overall, 43 (93%) of 46 patients had treatment-related adverse events during neoadjuvant treatment, with 30% being grade 3 or worse; however, none of the adverse events were associated with surgery delays or deaths. Based on this study and on the aforementioned LCMC3 trial [10], the combination strategies of neoadjuvant and adjuvant immunotherapy could represent a valid therapeutic option, providing the advantages of a synergistic immunotherapeutic effect in these patients. Neoadjuvant ICIs could elicit a sustained immune response by producing several changes in the tumor microenvironment, while adjuvant immunotherapy, exploiting this effect as well, could lead to a durable systemic control. However, further clinical trials directly comparing neoadjuvant immunotherapy to adjuvant or to neoadjuvant combined with adjuvant immunotherapy in resectable NSCLC are needed.

Another open-label, multicenter, single-arm, phase II study investigated the efficacy of atezolizumab in combination with carboplatin nab-paclitaxel in 30 patients with resectable stage IB-IIIA NSCLC [20]. Non-smoking patients were excluded from the enrollment.

Treatment was administered for four cycles. The primary endpoint was the proportion of MPR. Secondary endpoints included DFS and OS. An increase in the percentage of MPR from 22% to 44% thanks to the addition of atezolizumab was expected. Results demonstrated that 87% of the patients underwent R0 radical surgery. The percentage of MPR was 57%, with 33% pCR. The efficacy was independent of PD-L1 expression. The median DFS was 17.9 months, and the median OS was not reached. The molecular characterization was available in 13 cases. Three patients carried STK11 mutations, four carried EGFR mutations, one HER2 carried an mutation and two carried KRAS mutations. No partial response was observed in those harboring STK11 mutations, while pCR was evidenced in two patients with EGFR mutations (exon 21 L858R and L858R/S768I).

These findings were confirmed in the randomized, phase III CheckMate 816 study, which was designed to demonstrate the major efficacy of three cycles of chemotherapy plus nivolumab over chemotherapy alone in patients with resectable stage IB-IIIA NSCLC [21]. Patients carrying EGFR mutations or EML4-ALK translocations were excluded from the enrollment. Co-primary endpoints were event-free survival (EFS) and the percentage of pathological complete response, both evaluated by blinded independent review. Secondary endpoints included MPR, OS and time to death or distant metastases. Surgery was planned to occur within 6 weeks of the completion of neoadjuvant treatment; after which, patients in both groups could receive up to four cycles of adjuvant chemotherapy, radiotherapy or both. Combination therapy significantly prolonged EFS (31.6 months with nivolumab plus chemotherapy and 20.8 months with chemotherapy alone (HR 0.63, CI 0.43–0.91, $p = 0.005$)). At 1 and 2 years, the estimated percentage of patients surviving without disease progression or disease recurrence was 76.1% and 63.8% with nivolumab plus chemotherapy and 63.4% and 45.3% with chemotherapy alone. The EFS benefit with nivolumab plus chemotherapy was maintained after adjustment for adjuvant therapy, which was administered in 11.9% of the patients in the nivolumab plus chemotherapy group and 22.2% of those in the chemotherapy alone group. Of note, EFS across most key prespecified subgroups favored nivolumab and chemotherapy. However, the magnitude of benefit was greater in the subgroup of patients with stage IIIA NSCLC, in those with PD-L1 $\geq$ 1% and in those with non-squamous histology. The percentage of patients with a pCR was 24.0% with nivolumab plus chemotherapy and 2.2% with chemotherapy alone ($p < 0.001$). Significantly higher MPR were observed in the nivolumab plus chemotherapy arm over chemotherapy (36.9% versus 8.9%). The benefit of the combination arm on pathological response was observed regardless of PD-L1 expression. At the first prespecified interim analysis, OS data were not mature. Of note, the addition of nivolumab to neoadjuvant chemotherapy did not increase the incidence of AEs or impede the feasibility of surgery. Interestingly, in an exploratory analysis, EFS was longer in patients with a pCR than in those without, suggesting a promising role of pCR as an early indicator of therapeutic efficacy in resectable NSCLC. Additionally, the depth of pathological regression appeared to be predictive of improved EFS in the nivolumab and chemotherapy group [22]. Based on these results, the food and drug administration (FDA) approved nivolumab in combination with chemotherapy in patients with resectable NSCLC with tumors $\geq$ 4 cm or the involvement of the lymph nodes, thus becoming the new standard of care in the therapeutic landscape of neoadjuvant treatment strategies in patients with NSCLC.

An overview of the main phase II and phase III clinical trials exploring the efficacy of ICIs in the neoadjuvant setting is shown in Table 1.

**Table 1.** Neoadjuvant immunotherapy in NSCLC.

| Trial (ClinicalTrials.gov Identifier) | Phase | Stage | N. of pts | Therapy | Primary Endpoint(s) | Grade ≥ 3 TRAEs (%) | MPR (%) | OS (%) | R0 Surgery (%) |
|---|---|---|---|---|---|---|---|---|---|
| NCT02259621 | II | I–IIIA | 21 | Nivolumab for 2 cycles | Safety | 4–5 | 45 | 80 (at 5 years) | 95.2 |
| LCMC3 (NCT02927301) | II | IB–IIIB | 181 | Atezolizumab for 2 cycles | MPR | 16.60 | 20 | 80 (at 3 years) | 82.3 |
| IFCT-IONESCO (NCT03030131) | II | IB–IIIA | 46 | Durvalumab for 3 cycles | % of complete resection (R0) | 0 | 19 | 89 (at 12 months) | 89.1 |
| NEOMUN (NCT03197467) | II | II–IIIA | 30 | Pembrolizumab for 2 cycles | Safety, ORR | 30 | 27 | NR | 100 (interim analysis on 15 pts) |
| NEOSTAR (NCT03158129) | II | IA–IIIA | 44 | • Experimental Arm A: nivolumab on days 1, 15 and 29 • Experimental Arm B: nivolumab as in Arm A + ipilimumab on day 1 | MPR | • Arm A: 4.3 • Arm B: 4.8 | • Arm A: 22 • Arm B: 38 | NR | • Arm A: 95.7 • Arm B: 81.0 |
| NADIM (NCT03081689) | II | IIIA | 46 | 3 cycles of nivolumab plus paclitaxel plus carboplatin → adjuvant Nivolumab for 1 year | 24 m-PFS | 30.4 | 82.9 | 81.9 (at 36 months) 78.9 (at 42 months) | 89.1 |
| CheckMate816 (NCT02998528) | III | IB–IIIA | 358 | • Experimental arm: nivolumab (360 mg) + platinum-doublet chemotherapy • Active comparator: platinum-doublet chemotherapy alone for three cycles | EFS, % of pCR | NE | • Exp arm: 36.9 • Active comparator: 8.9 | NR | • Exp arm: 69.3 • Active comparator: 58.7 |
| TOP1501 (NCT02818920) | II | IB–IIIA | 35 | Neoadjuvant pembrolizumab × 2 cycles → adjuvant chemotherapy × 4 cycles → adjuvant pembrolizumab × 4 cycles | Surgical Feasibility Rate | 0.35 | 28 | NE | 88 |

EFS, event-free survival; MPR, major pathological response; NE, not evaluated; NR, not reached; ORR, objective response rate; OS, overall survival; PFS, progression-free survival; R0, R0 resection indicates a microscopically margin-negative resection, in which no gross or microscopic tumor remains in the primary tumor bed; TRAEs, treatment-related adverse events.

## 5. ICIs in the Adjuvant Setting

For more than 20 years, platinum-doublet chemotherapy has represented the standard treatment in the adjuvant setting, with unsatisfactory results in terms of OS. Based on positive efficacy results with the use of ICIs in the metastatic setting, these agents have been tested in the curative setting in order to improve clinical outcomes. Results from the phase III IMpower-010 trial demonstrated the efficacy of the use of ICIs as an adjuvant strategy in resected NSCLC patients. This multicenter, open-label trial randomized 1005 patients with stage 1B (tumors $\geq$ 4 cm) to IIIA NSCLC between atezolizumab, administered for one year, or best supportive care (BSC), following one to four cycles of platinum-doublet chemotherapy [23]. This study was designed to demonstrate an improvement in terms of DFS in patients receiving atezolizumab. Results showed a significant benefit in patients with stage II-IIIA and PD-L1 $\geq$ 1%, as assessed by the Ventana (SP263) assay, and in all patients with stage II-IIIA NSCLC, irrespective of PD-L1 expression. In this study, the benefit was greater in the subgroup with PD-L1 $\geq$ 1%. A smaller but statistically significant benefit was also found in the intent to treat population (stage IB–IIIA). Based on these findings, adjuvant atezolizumab was first FDA-approved in patients with stage IB-IIIA and PD-L1 $\geq$ 1%, without sensitizing EGFR mutations, following cisplatin-based chemotherapy [24]. Conversely, EMA has approved adjuvant atezolizumab in patients with stage IB-IIIA NSCLC, without sensitizing EGFR mutations, following cisplatin-based chemotherapy, with PD-L1 expression $\geq$ 50%, based on the results of the secondary endpoints of the IMpower010 trial that included DFS in patients with stage II-IIIA tumors expressing PD-L1 on 50% or more tumor cells. Indeed, this subgroup of patients showed a significative improvement in terms of DFS with atezolizumab compared to BSC (median NE vs. 35.7 months, HR: 0.43, 95% CI 0.27–0.68) [23,25] (www.ema.europa.eu, accessed on 3 January 2023).

Similar results were found in the phase III PEARLS/KEYNOTE-091, which was designed to evaluate the efficacy of pembrolizumab in completely resected, pathologically confirmed stage IB (tumors of $\geq$4 cm in diameter), II or IIIA NSCLC. Conversely to IMpower-010, adjuvant chemotherapy was not mandatory, but to be considered for stage IB and strongly recommended for stage II and IIIA NSCLC. Overall, 1177 patients, stratified by disease stage, previous adjuvant chemotherapy, PD-L1 expression and geographical region, were randomized to receive pembrolizumab or placebo for up to 18 cycles. Co-primary endpoints were DFS in the overall population and in the population with PD-L1 $\geq$ 50%. Pembrolizumab significantly prolonged DFS in the overall population. Results from the subgroup with PD-L1 $\geq$ 50% are still not mature [26].

An overview of the clinical trials exploring the efficacy of ICIs in the adjuvant setting is shown in Table 2.

**Table 2.** Adjuvant immunotherapy in NSCLC.

| Trial (ClinicalTrials.gov Identifier) | Phase | Stage | N. of pts | Therapy | Primary Endpoint | mDFS | Secondary Endpoint(s) | mOS | Grade $\geq$ 3 TRAEs |
|---|---|---|---|---|---|---|---|---|---|
| IMPOWER010 (NCT02486718) | III | IB–IIIA | 1005 | Atezolizumab for 1 year vs. BSC | DFS | NR vs. 37.2 months (HR 0.81 95% CI) | OS; safety | NR [†] | 22% vs. 11.5% |
| PEARLS/KEYNOTE-091 (NCT02504372) | III | IB–IIIA | 1177 | Pembrolizumab vs. placebo for 1 year | DFS | 53.6 vs. 42 months (HR 0.76; 95% CI) | OS | NR * | 34.1% vs. 25.8% |

[†] preliminary stratified HR 0.995 (95% CI, 0.78–1.28). * preliminary HR 0.87 (95%, 0.67–1.15); the significance boundary for OS in the all-comers population was not crossed (18-mo rate 91.7% vs. 91.3%). BSC, best supportive care; CI, confidence interval; DFS, disease-free survival; HR, hazard ratio; mDFS, median disease-free survival; NR, not reached; OS overall survival; TRAEs, treatment-related adverse events.

In addition, there are several ongoing clinical trials evaluating the role of adjuvant immunotherapy in patients undergoing surgery for earlier-stage NSCLC. Among these, the phase II BTCRC-LUN18-153 trial (NCT04317534) was designed to evaluate whether administering pembrolizumab once every four weeks following surgical resection for up to nine cycles improves DFS compared to observation following surgical resection in patients

with primary tumors measuring less than 4 cm (stage I) (www.clinicaltrials.gov, accessed on 10 March 2023).

Other clinical trials, exploring the efficacy of ICIs as neoadjuvant/adjuvant strategies, are currently ongoing, as shown in Table 3.

**Table 3.** Current ongoing clinical trials exploring the role of neoadjuvant or adjuvant ICIs in NSCLC patients.

| Neoadjuvant | | | | |
|---|---|---|---|---|
| **Trial (ClinicalTrials.gov Identifier)** | **Phase** | **Stage** | **Treatment** | **End Points** |
| KEYNOTE -671 (NCT03425643) | III | II–IIIA, resectable IIIB | Experimental arm: pembrolizumab and chemotherapy (cisplatin + gemcitabine/pemetrexed) × 4 cycles → adjuvant pembrolizumab × 13 cycles Comparator arm: placebo and chemo × 4 cycles→ adjuvant placebo | EFS, OS |
| CheckMate 77T (NCT04025879) | III | II–IIIB | Experimental arm: Neoadjuvant nivolumab + platinum-based doublet chemotherapy × 4 cycles → adjuvant nivolumab for 1 year Comparator arm: Neoadjuvant Placebo + platinum-based doublet chemotherapy × 4 cycles → adjuvant placebo | EFS |
| IMpower030 (NCT03456063) | III | II–IIIB | Experimental arm: atezolizumab + platinum-based chemotherapy × 4 cycles → adjuvant atezolizumab × 16 cycles Placebo Comparator: placebo + platinum-based chemotherapy × 4 cycles → adjuvant placebo | EFS |
| AEGEAN (NCT03800134) | III | II–IIIB | Experimental arm: durvalumab + platinum-based chemotherapy × 4 cycles Placebo Comparator: placebo + platinum-based chemotherapy × 4 cycles | EFS, pCR |
| NEOpredict (NCT04205552) | II | IB–IIIA | Nivolumab or nivolumab/relatlimab × 2 cycles | Feasibility |
| Adjuvant | | | | |
| **Trial (ClinicalTrials.gov Identifier)** | **Phase** | **Stage** | **Treatment** | **End Points** |
| ANVIL (NCT02595944) | III | IB–IIIA | After adjuvant chemotherapy → Nivolumab for 1 year vs. BSC | DFS, OS |
| BR.31 (NCT02273375) | III | IB–IIIA | Experimental arm: after adjuvant chemo (if appropriate) → durvalumab for 1 year Comparator arm: after adjuvant chemotherapy (if appropriate) → placebo for 1 year | DFS |
| MERMAID-1 (NCT04385368) | III | II–III (MRD+) | Experimental arm: durvalumab + SoC chemotherapy Placebo Comparator: placebo + SoC chemotherapy | DFS (MRD+) |
| MERMAID-2 (NCT04642469) | III | II–III (MRD+) | Experimental arm: after adjuvant chemotherapy (if appropriate) → (if MRD1 within 96 weeks post surgery) durvalumab for 2 years Placebo comparator: after adjuvant chemotherapy (if appropriate) → (if MRD1 within 96 wk post surgery) placebo for 2 years | DFS in PD-L1 TPS ≥ 1% |
| NADIM-ADJUVANT (NCT04564157) | III | IB–IIIA | Experimental arm: carboplatin/Paclitaxel × 4 cycles → nivolumab × 6 cycles Comparator arm: carboplatin/paclitaxel × 4 cycles → BSC | DFS |
| BTCRC-LUN18-153 (NCT04317534) | II | I | Experimental arm: Pembrolizumab 400 mg IV every 6 weeks × 9 cycles Comparator arm: Observation | DFS |

EFS, event-free survival; OS, overall survival; MPR, major pathological response; DFS, disease-free survival; MRD, minimal residual disease; pCR, pathological complete response; BSC, best supportive care.

There are also some studies that are evaluating the association between immunotherapy and stereotactic body radiotherapy (SBRT) in early-stage NSCLC patients. In the phase III trial Keynote-867 (NCT03924869), approximately 530 patients with stage I/II NSCLC are randomized 1:1 to receive SBRT and either pembrolizumab 200 mg or placebo every 3 weeks for 17 cycles (approximately 1 year) until disease recurrence or development of unacceptable toxicity. The primary endpoints are EFS by blinded indipendent central review (BICR) and OS, and the study is still ongoing. The ongoing phase III trial PACIFIC-4 RTOG 3515 (NCT03833154) has been designed to assess the efficacy and safety of durvalumab with SBRT versus placebo with SBRT in patients with unresected clinical stage I/II lymph node-negative (T1 to T3N0M0) NSCLC. The primary endpoint is PFS; other endpoints include OS and safety. Another ongoing phase III study, SWOG/NRG S1914 (NCT04214262), is evaluating the association of SBRT with atezolizumab in patients with T1-3N0M0 NSCLC who are medically inoperable or decline surgery. Patients are randomized 1:1 to standard-of-care SBRT or to neo-adjuvant, concurrent and adjuvant atezolizumab 1200 mg IV Q3 weekly with SBRT initiated with cycle 3. The primary endpoint is OS, secondary endpoints are PFS and safety (www.clinicaltrials.gov, accessed on 10 March 2023).

## 6. Biomarkers for Patient Selection

Despite the significant progress that has been made in terms of pathological responses with the introduction of ICIs in the neoadjuvant setting, chemo-immunotherapy remains ineffective in a significant proportion of patients (40%). The molecular and immunological bases responsible for sensitivity or resistance remain to be fully determined; however, emerging biomarkers have been described. In order to identify molecular markers for patient selection, tumor samples from patients enrolled in the NADIM study were analyzed [27]. Results showed a pro-inflammatory gene expression profile, a higher T cell receptor repertoire clonality and the upregulation of genes involved in the signaling pathway of IFN-γ in pretreatment tissues of those patients obtaining pCR. Dendritic cells, T helper cells, cytotoxic T cells and NK cells favor the release of IFN-γ, which, once activated, increases the expression of MHC class molecules, activates NK and T cells and promotes the switching of macrophages to M1 macrophages, thus enhancing immune activation. Conversely, the upregulation of genes involved in proliferation was evidenced in tissue from patients with non-pCR. Another exploratory analysis in blood from patients enrolled in the NADIM trial showed the presence of differences in terms of cytotoxic profile and peripheral blood mononuclear cells (PBMCs) in patients with complete- and non-pCR [28]. The PD-1 expression was evaluated in pretreatment peripheral blood CD4+ T cells, CD8+ T cells, CD5+ T cells and NK cells. Higher levels of PD-1+ cells and a lower expression of CTLA-4 on monocytes were found in patients achieving complete pathologic remission.

Moreover, in the NADIM trial, baseline circulating tumor DNA (ctDNA) levels were significantly associated with tumor size. Patients characterized by ctDNA levels < 1% mutant allele fraction (MAF) at baseline showed significantly improved PFS and OS compared to patients with increased ctDNA levels. In addition, further improvements in PFS and OS were found in patients with undetectable ctDNA after neoadjuvant treatment [19]. Likewise, similar results were shown in the prior study, CheckMate 816, in which increased ctDNA clearance was detected in patients treated with nivolumab plus chemotherapy compared to those treated with chemotherapy alone. However, a high percentage of pCR was obtained in patients with increased ctDNA clearance in both treatment groups compared with those without ctDNA clearance [21].

These data suggest the use of blood-based biomarkers to identify those patients who could benefit more from neoadjuvant chemoimmunotherapy. Further studies are needed to confirm these findings.

## 7. Comments and Future Perspectives

ICIs have been demonstrated to play a meaningful role in resectable patients with NSCLC. Data from preclinical studies demonstrated the major efficacy of ICI therapy in

the prolongation of OS when administered in the neoadjuvant compared with adjuvant setting due to the higher probability of inducing peripheral tumor-specific immune responses. Indeed, neoadjuvant immunotherapy provides an effective strategy to early treat micrometastatic disease and enhances the immune response when bulk tumor and tumor antigens are still present during treatment. Results from phase II and phase III clinical trials have shown that immunotherapy or chemo-immunotherapy or dual ICIs significantly improve the percentage of pCR and MPR. It is important to underline that, across all studies, there was high heterogeneity in the selection of primary endpoints, which included EFS, DFS or PFS. However, the pathologic assessment of response has been incorporated into all the trials, since it can be considered a surrogate endpoint of survival benefit, as previously demonstrated in different studies, including CheckMate 816 [22]. Indeed, a meta-analysis including data from patients with NSCLC enrolled in clinical trials exploring the efficacy of neoadjuvant chemotherapy or chemoradiotherapy, showed a significant association between pathologic response and EFS and OS, thus confirming the efficacy of this endpoint to assess the clinical benefit from neoadjuvant strategies [29].

Open questions remain about the scoring approaches to be used in the clinical practice for pathological response assessments to neoadjuvant immunotherapy in NSCLC, the optimal duration of neoadjuvant/adjuvant therapy, the role of adjuvant treatment following neoadjuvant strategies, the identification of plasma or tissue biomarkers to select those patients who could benefit more from ICIs and the efficacy of targeting PD-1 or PD-L1 in patients carrying targetable molecular alterations, including EGFR mutations, HER2 mutations and EML4-ALK, ROS1, NTRK or RET rearrangements. There is a need to define a diagnostic methodology that would yield reproducible information of clinical value. In patients with metastatic NSLC, PD-L1 still represents the only validated biomarker used to select immunotherapy. Discordant results on the role of PD-L1 expression have been found in trials with ICIs in the early stages of disease. In some cases, the efficacy was independent of PD-L1 status [20], while in CheckMate 816, the efficacy of chemo-immunotherapy was higher in patients with PD-L1 $\geq$ 1% [21]. There was heterogeneity in the inclusion criteria of the different studies, as in some cases patients with EGFR mutations or EML4-ALK rearrangement were excluded, while others did not allow the enrollment of non-smoker patients, but molecular characterization was not mandatory to enter the trial. Few molecular data are available, but there was concordance about the poor efficacy of ICIs in patients carrying STK11 mutations.

Finally, novel strategies to monitor tumor recurrence are required. Circulating tumor DNA (ctDNA) has been extensively used in metastatic patients, and recently its prognostic and predictive value has been evaluated in the early stages. Its quantification might indicate a non-response to neoadjuvant therapy or its presence might be the signal of an early recurrence. This could be particularly useful in cases developing nodal immune flare, in order to distinguish pseudo-progression from progression.

**Author Contributions:** C.L., C.C.S., G.C., S.M., M.D.P., A.S. and M.I.P. conducted the literature search for the review. C.L., C.C.S., G.C., V.G. and M.S. wrote the manuscript. C.L., V.G. and M.S. supervised the project and contributed to the final version of the manuscript. All authors have read and agreed to the published version of the manuscript.

**Funding:** This research received no external funding.

**Conflicts of Interest:** The authors declare no conflict of interest.

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
