# Peer review of "Immunotherapy in Early-Stage Non-Small Cell Lung Cancer (NSCLC): Current Evidence and Perspectives"

_curroncol, doi:10.3390/curroncol30040280_

Round 1

Reviewer 1 Report

The authors did an outstanding job with this review.  The review thoughtfully discussed the role of immunotherapy in earlier stage non small cell lung cancer.  I have only a few minor suggestions:

1) can the authors briefly discuss and highlight that there are now 3 major randomized trials evaluating immunotherapy and SBRT for stage IA NSCLC.

2) can the authors discuss that there is an ongoing study of adjuvant immunotherapy in patients who undergo surgery for stage IA NSCLC.

https://bigtencrc.org/big-ten-crc-study-tests-pembrolizumab-vs-observation-following-surgery-in-stage-i-non-small-cell-lung-cancer/

Author Response

Point 1.  Can the authors briefly discuss and highlight that there are now 3 major randomized trials evaluating immunotherapy and SBRT for stage IA NSCLC.

Response 1. Thank you for the comment. We have added a brief description about the 3 studies evaluating combination of SBRT and immunotherapy in stage IA NSCLC (see lines 355-370).

Point 2. Can the authors discuss that there is an ongoing study of adjuvant immunotherapy in patients who undergo surgery for stage IA NSCLC. 

https://bigtencrc.org/big-ten-crc-study-tests-pembrolizumab-vs-observation-following-surgery-in-stage-i-non-small-cell-lung-cancer/

Response 2.  Thank you for your comment. We have mentioned this ongoing study (see lines 341-346) and we have also added it to table 3.

Reviewer 2 Report

In this manuscript, the authors provide a nice review of neoadjuvant and adjuvant immune checkpoint inhibitor (ICI) therapies in treating non-small cell lung cancer (NSCLC). Yet, for the reader’s ease of understanding, the following additions are recommended before publication.

It would be super helpful for the general audience in the cancer field if the neoadjuvant and adjuvant settings could be clearly defined in the Introduction section. Also, about the paragraph starting in line 195, shall the treatment regimen tested in NADIM represent a combination of neoadjuvant and adjuvant settings, since “[f]ollowing surgery, patients received adjuvant nivolumab for one year”?  Similar comments to LCMC3 (NCT02927301) listed in Table 1, in view of the clinicaltrials.gov disclosure stating “[f]llowing surgery, adjuvant therapy consisted of up to 12 months of atezolizumab in participants who demonstrate clinical benefit with neoadjuvant therapy.” Does combining neoadjuvant ICI with adjuvant ICI provide any advantages and justify additional discussion?

Along this line, shall we include “resectable NSCLC” and/or “adjuvant and neoadjuvant immunotherapy” in the title? If not, we may like to include discussions on more clinical trials, such as KEYNOTE-789 (see, https://www.clinicaltrials.gov/ct2/show/NCT03515837 and https://www.merck.com/news/merck-provides-update-on-phase-3-trials-keynote-641-and-keynote-789/).

With respect to TOP1501 identified in Table 3, some data has been made available. See, for example, https://pubmed.ncbi.nlm.nih.gov/33985811/. Further in Table 3, clinicaltrials.gov identifiers are needed.

In addition, the following minor changes would be really appreciated.

(1)         Full names of acronyms need to be provided on their first appearances, such as NSCLC in the title line and ICI.

(2)         The cell receptors illustrated in the center of Figure 1 seem detached from the cells.

(3)         Is the term “perioperative” in the subsection title spanning lines 72-73 suitable? It seems the mechanism review in this subsection only focuses on the neoadjuvant setting.

Author Response

  • It would be super helpful for the general audience in the cancer field if the neoadjuvant and adjuvant settings could be clearly defined in the Introduction section.

Response 1.  Thank you for your comment. We provided to insert a definition of adjuvant setting in lines 42-48, while the neoadjuvant setting was already described in lines 58-61 and we also tried to make it clearer.

  • About the paragraph starting in line 195, shall the treatment regimen tested in NADIM represent a combination of neoadjuvant and adjuvant settings, since “[f]ollowing surgery, patients received adjuvant nivolumab for one year”? Similar comments to LCMC3 (NCT02927301) listed in Table 1, in view of the clinicaltrials.gov disclosure stating “[f]llowing surgery, adjuvant therapy consisted of up to 12 months of atezolizumab in participants who demonstrate clinical benefit with neoadjuvant therapy.” Does combining neoadjuvant ICI with adjuvant ICI provide any advantages and justify additional discussion?

Response 2. Thank you for your comment. We have included some comments about this issue in the paragraph "Combination strategies in the neoadjuvant setting". Combining neoadjuvant ICI with adjuvant ICI could represent a valid therapeutic option for resectable NSCLC. However, further clinical trials are needed.  

  • Along this line, shall we include “resectable NSCLC” and/or “adjuvant and neoadjuvant immunotherapy” in the title? If not, we may like to include discussions on more clinical trials, such as KEYNOTE-789 (see, https://www.clinicaltrials.gov/ct2/show/NCT03515837and https://www.merck.com/news/merck-provides-update-on-phase-3-trials-keynote-641-and-keynote-789/)

Response 3. Thank you for the comment. We have chosen the title “immunotherapy in early stage NSCLC (…)” because in this review we discuss about resectable and non-resectable early stage non-small cell lung cancer, but not about the role of immunotherapy in metastatic disease.

  • With respect to TOP1501 identified in Table 3, some data has been made available. See, for example, https://pubmed.ncbi.nlm.nih.gov/33985811/. Further in Table 3, clinicaltrials.gov identifiers are needed.

Response 4. Thank you for your comment. We provided to insert TOP1501 and relative available data in the Table 2. We also have added clinicaltrial.gov identifiers in the Table 3.

In addition, the following minor changes would be really appreciated.

  • Full names of acronyms need to be provided on their first appearances, such as NSCLC in the title line and ICI.

Response 1.  Thank you for your comment. We have revised the acronyms providing the full name on their first appearance.

  • The cell receptors illustrated in the center of Figure 1 seem detached from the cells.

Response 2. Thank you. We have modified Figure 1.

  • Is the term “perioperative” in the subsection title spanning lines 72-73 suitable? It seems the mechanism review in this subsection only focuses on the neoadjuvant setting

Response 3. Thank you for your comment. We changed the term to “preoperative” .